# Robust Tracker of Hybrid Microgrids by the Invariant-Ellipsoid Set

Hilmy Awad [1], Ehab H. E. Bayoumi [2,*], Hisham M. Soliman [3] and Michele De Santis [4]

1 Department of Electrical Technology, Faculty of Technology and Education, Helwan University, El-Sawah, Cairo 11813, Egypt; hilmy_awad@yahoo.com
2 Energy and Renewable Energy Department, Egyptian Chinese University, Cairo 11724, Egypt
3 Department of Electrical Power Engineering, Faculty of Engineering, Cairo University, Cairo 12613, Egypt; hsoliman1@yahoo.com
4 Department of Engineering, Niccolò Cusano University, 00166 Roma, Italy; michele.desantis@unicusano.it
* Correspondence: ehab.bayoumi@gmail.com

**Abstract:** This paper introduces a new ellipsoidal-based tracker design to control a grid-connected hybrid direct current/alternating current (DC/AC) microgrid (MG). The proposed controller is robust against both parameters and load variations. The studied hybrid MG is modelled as a nonlinear dynamical system. A linearized model around an operating point is developed. The parameter changes are modelled as norm-bounded uncertainties. We apply the new extended version of the attractive (or invariant) ellipsoid for this tracking problem. Convex optimization is used to obtain the region's minimal size where the tracking error between the state trajectories and the reference states converges. The sufficient conditions for stability are derived and solved based on linear matrix inequalities (LMIs). The proposed controller's validity is shown via simulating the hybrid MG with various operational scenarios. In each scenario, the performance of the controller is compared with a recently proposed sliding mode controller. The comparison clearly illustrates the superiority of the developed controller in terms of transient and steady-state responses.

**Keywords:** convex optimization; ellipsoidal design; hybrid microgrids; LMI; robust trackers





## 1. Introduction

Technology advances in power generation, control, computer hardware, and software have led to the spread of microgrids (MGs). Industrial customers aiming to reduce the cost and increase their reliability and resilience have driven the MGs' evolution. As a result, the MGs are currently the primary power source for many commercial and residential applications [1].

Locality, independence, and intelligence are the key characteristics of the MG. The MG's locality provides an optimal solution to the power losses issue of a typical power system structure, reaching up to 40% of the generated power. In many cases, the MG is the only solution to the power supply of remote areas where the public grid is inaccessible or does not exist [2].

MGs' independence is attributed to their ability to supply the total load demand in the outage of the utility services or the existence of severe disturbances. The MG can operate synchronously with the utility grid or in isolated mode. When isolated, it can completely control its voltage and frequency, thanks to the dedicated power converters and their advanced control schemes [3].

Embedded sensors, communications channels, innovative software, and powerful controllers have instigated MG's intelligence. Intelligence in this context implies the capability of self-monitoring, evaluation, and decision-making to optimally utilize and operate the available energy resources in addition to the energy-management system [4].

MG's installation and commissioning are soaring. For instance, in 2019 (second half only), there were 4475 projects adding up to 27 GW of installed capacity [5]. California State, USA, has demonstrated great impetus to obtain 50% of its electricity via MGs

and renewable sources by 2030 [6]. MGs' development in the European Union Nations, Japan, Korea, Australia, and North America is reported in [7], showing explicitly that the implementations are growing rapidly.

Based on the integration level into the primary grid, the MGs are classified into a remote-, facility- and utility-MG, ordered by their impact on the power system from low to high. The MG can be a direct current (DC), alternating current (AC), or hybrid system. Figure 1a displays a layout example of an AC MG, and Figure 1b depicts an outline of a DC MG. Statistically, the AC-MG system configuration is still dominant due to mainly two factors: AC-load availability and ease of protection [8]. However, some factors push the DC MG to compete with the AC MG, such as the DC-based renewable system's spread and ease of control. A hybrid MG aims to merge the merits of both systems [9,10]. Figure 2 displays one possible layout of a hybrid MG.

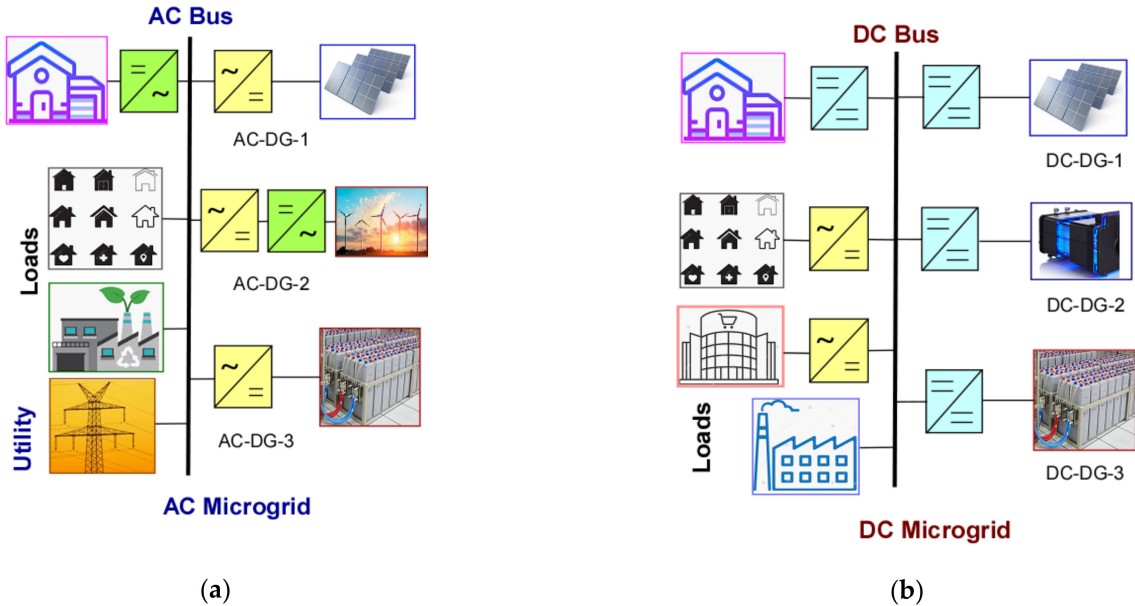

**Figure 1.** Layouts of alternating current (AC) microgrid (MG) (**a**) and direct current (DC) MG (**b**).

The hybrid MG provides a comprehensive solution to common issues like the suitability of sources and loads. It can integrate both DC and AC types of energy source and customer loads with the minimal (or zero) conversion stages. As shown in Figure 2, the interlinking inverter facilitates the power flow between the DC and AC MG and the main utility via the appropriate controllers. Thus, a hybrid MG's autonomous operational capability is significantly higher than only DC- or AC-MG.

The MG's control is either centralized or decentralized. Centralized control systems have the following advantages: (1) the main objectives are identified and realized; (2) they can offer global optimal decisions/solutions; (3) their synchronization to the main utility is relatively easy, and they can efficiently be operated online. On the other hand, decentralized control systems have the advantages: (1) they are appropriate for quickly changing infrastructures; (2) they can be easily expanded due to high capabilities of plug and play operation; (3) their reliability is high; (4) their communication and computational cost are relatively low [11,12]. Both centralized and decentralized controllers have been realized by many control algorithms such as $H\infty$ [13], LQR [14], sliding mode [15], model-predictive control [16] and artificial intelligence techniques including fuzzy logic [5,17], neural networks [6], and genetic algorithms [18].

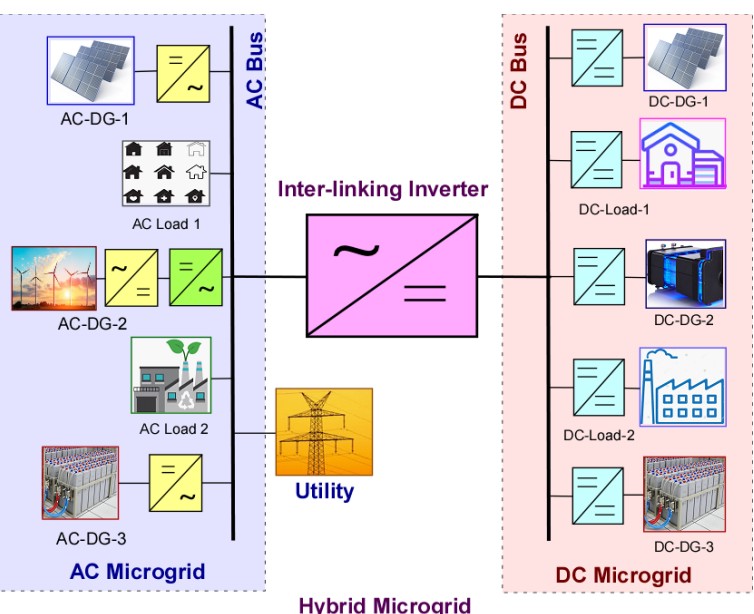

**Figure 2.** Layouts of hybrid MG.

**Remark 1.** *It should be emphasized that [14] solves the problem of severe transients that occur during the transition between the connected and islanded modes of microgrids (which affect the voltage and frequency responses). The seamless transition is achieved by the linear quadratic regulators LQR with a simple pole placement method to select the weighting matrices Q and R. The bumpless transition is not considered in the present manuscript. The adaptive sliding control given in [15] achieves fast bus voltage tracking without additional sensors in addition to system scalability and maintaining the plug-and-play operation of the Distribution Generation units (DGs). By using the proposed method [15] to estimate the uncertainty and external disturbance, the system parameters and switching function can be adjusted in time to weaken the chattering. However, the control of [15] is non-linear, has some chattering, and is applied to only DC microgrids. The model-predictive control [16] is a time-varying tracker and requires significant computational time. The methods [5,6,18] also require extensive training and computational burdens.*

Furthermore, decentralized control techniques manage, organize, and control an efficient short alteration from grid-connected mode to islanded mode [19]. It introduces and allows recent interlinking converter circuits to be used as quasi-Z-source converters [19]. The interlinking converter could be connected in series or parallel to enhance the hybrid MGs' power quality. The series interlinking converter is used to improve the hybrid MGs, and the parallel voltage quality interlinking converter compensates for the harmonic voltage generated by the non-linear loads. In [20], a new circuit combination is anticipated for the interlinking converter to improve power quality indices. The interlinking converter is divided into two parts: parallel and series interlinking converter.

This paper introduces a control algorithm using invariant ellipsoids. The presented control technique avoids the complexity of selecting the weights of $H\infty$ and the chattering issues of the sliding mode control. Moreover, it handles the parameter's uncertainties and loads variations effectively. The contribution of the paper is:

(1) A novel theorem, based on the linear matrix inequalities, is introduced to guarantee the optimal trackers' solvability, in which the outputs follow the desired inputs and achieve external disturbance rejection.

(2) A new simple controller is designed based on the invariant sets (approximated and bounded by invariant ellipsoids) is introduced. The proposed controller is fast, with a very low overshoot, and there is no chattering, unlike the sliding mode approach. It is also robust against load changes and system parameters uncertainties.

(3)　The need to add a feedforward path (to enhance the DGs' controllers' performance) is eliminated [21].

The paper is structured as follows. Section 2 presents the model of the investigated hybrid MG and the statement of the research problem. In Section 3, the control design is presented and described in detail. The results are displayed in Section 4. Finally, the conclusions are presented in Section 5.

**Facts**

**Fact1:**

$$\mathbf{M}\Delta(t)\mathbf{N} + * < \epsilon\mathbf{M}\mathbf{M}' + \epsilon^{-1}\mathbf{N}'\mathbf{N}$$

The time-varying uncertainty $\Delta(t)$ can be removed using this fact

**Fact 2 (Schur complements)**

For a matrix $\mathbf{M}$ composed of

$$\mathbf{M} = \begin{bmatrix} \mathbf{M}_1 & \mathbf{M}_3 \\ * & \mathbf{M}_2 \end{bmatrix}$$

where $\mathbf{M}_1 = \mathbf{M}_1'$, $\mathbf{M}_2 = \mathbf{M}_2' > 0$ implies that $\mathbf{M} > 0$ if and only if

$$\mathbf{M}_1 - \mathbf{M}_3\mathbf{M}_2^{-1}\mathbf{M}_3' > 0$$

The last nonlinear matrix inequality can be linearized using this fact.

## 2. Model of the Hybrid Microgrid (MG) and Problem Statement

The model used in this paper is adapted from reference [19]. The studied hybrid MG consists of two DC sources, DC-DG1 and DC-DG2, as displayed in Figure 3. There are two AC sources, AC-DG1 and AC-DG2, and an inter-linking inverter. Details of the sources and inverters are depicted in Figure 4. DC-DG1 is a Photovoltaic (PV) system driven by a boost converter, as illustrated in Figure 4a. DC-DG2 is a Lithium-ion battery operated by a two-quadrant buck-boost converter. The AC-DG1 and AC-DG2 are PV and wind-turbine systems, respectively. They are controlled by three-phase voltage source converters, as shown in Figure 4b. Also, the inter-linking inverter is realized by a voltage source converter (Figure 4c). AC sources and the inter-linking converter are linked via Resistor-Inductor (RL) series filters. Referring to Figure 5, the AC side can be described by Equation (1).

$$
\begin{aligned}
L_{\text{ac}}\tfrac{d}{dt}i_{\text{a}} &= -R_{\text{ac}}i_{\text{a}} - v_{\text{a}} + u_{\text{a}}v_{\text{dc}} \\
L_{\text{ac}}\tfrac{d}{dt}i_{\text{b}} &= -R_{\text{ac}}i_{\text{b}} - v_{\text{b}} + u_{\text{b}}v_{\text{dc}} \\
L_{\text{ac}}\tfrac{d}{dt}i_{\text{c}} &= -R_{\text{ac}}i_{\text{c}} - v_{\text{c}} + u_{\text{c}}v_{\text{dc}}
\end{aligned}
\tag{1}
$$

where $u_{\text{a}}$, $u_{\text{b}}$, and $u_{\text{c}}$ are the modulation indices for three-phase a, b, and c, respectively. Transforming (1) to the dq-reference frame yields Equations (2) and (3).

$$
L_{\text{ac}}\frac{d}{dt}i_{\text{d}} = -R_{\text{ac}}i_{\text{d}} + \omega L_{\text{ac}}i_{\text{q}} - v_{\text{d}} + u_{\text{d}}v_{\text{dc}}
\tag{2}
$$

$$
L_{\text{ac}}\frac{d}{dt}i_{\text{q}} = -\omega L_{\text{ac}}i_{\text{d}} - R_{\text{ac}}i_{\text{q}} - v_{\text{q}} + u_{\text{q}}v_{\text{dc}}
\tag{3}
$$

$u_{\text{d}}$ and $u_{\text{q}}$, are d and the q-components of the modulation index of the inter-linking inverter, respectively. For the DC-DG1 and DC-DG2 sources, the following equations apply.

$$
C_{\text{s}}\frac{d}{dt}v_{\text{pv}} = -i_{\text{L1}} + i_{\text{pv}}
\tag{4}
$$

$$
L_1\frac{d}{dt}i_{\text{L1}} = -v_{\text{dc}}(1 - u_1) + v_{\text{pv}}
\tag{5}
$$

$$
L_2\frac{d}{dt}i_{\text{L2}} = -u_2v_{\text{dc}} + v_{\text{bat}}
\tag{6}
$$

$u_1$ and $u_2$, are the switching functions of DC-DG1 and DC-DG2, defined as follows.

$$
u_1 = \begin{cases} 1 & S_1 \text{ is on} \\ 0 & S_1 \text{ is off} \end{cases} \quad \text{and } u_2 = \begin{cases} 1 & S_2 \text{ is on} \\ 0 & S_2 \text{ is off} \end{cases}
$$

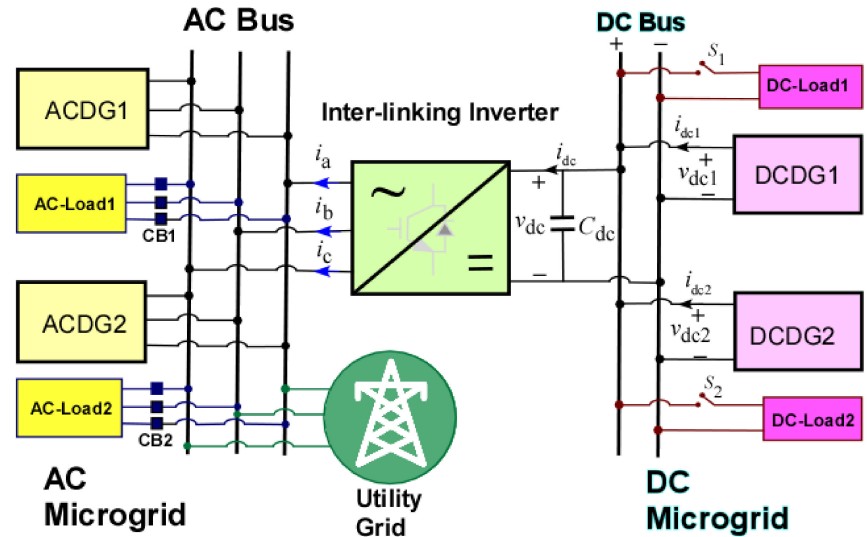

**Figure 3.** The layout of modelled hybrid MG.

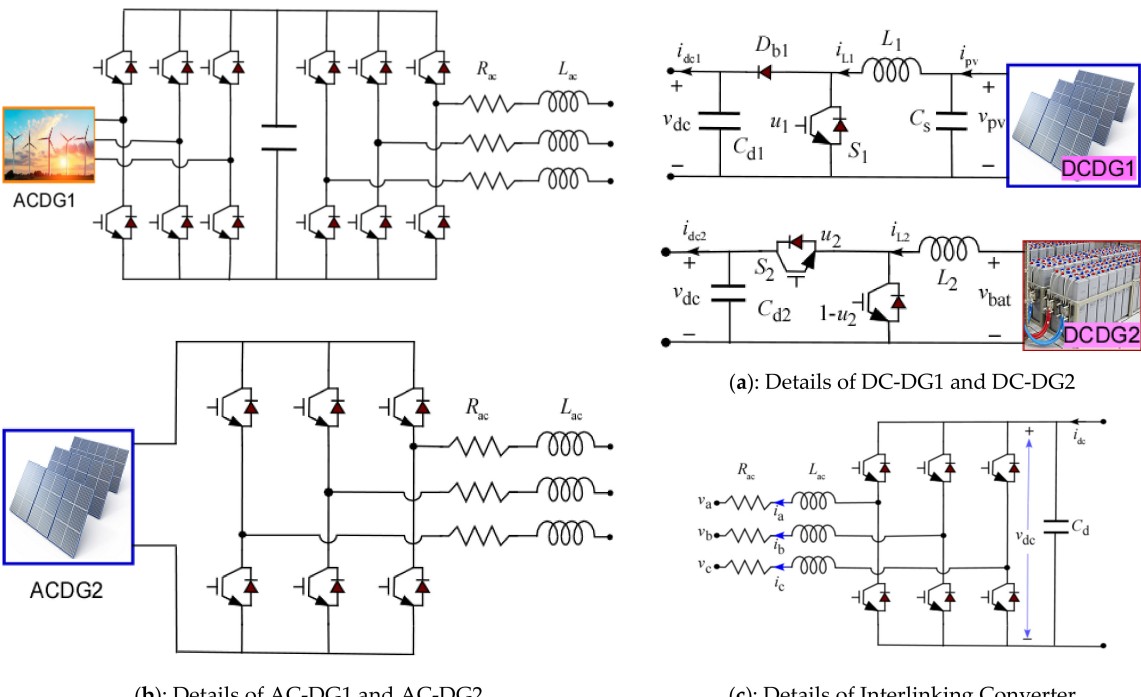

(**a**): Details of DC-DG1 and DC-DG2

(**b**): Details of AC-DG1 and AC-DG2

(**c**): Details of Interlinking Converter

**Figure 4.** Details of hybrid MG.

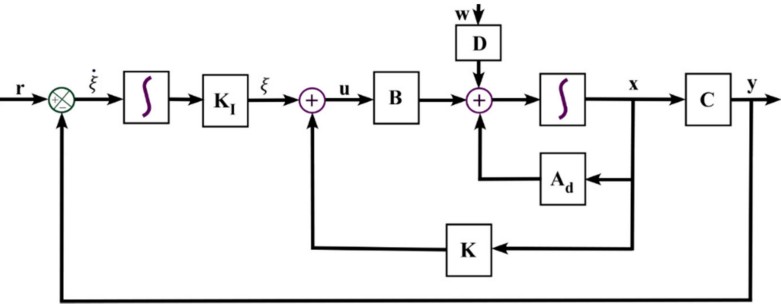

**Figure 5.** Block diagram of the control system.

The DC-link voltage changes according to the difference between the ac-side powers (currents) and the dc-side powers (currents). Looking from DC-DG1, the $v_{dc}$ state is expressed in Equation (7).

$$C_{d1}\frac{d}{dt}v_{dc} = i_{L1} - u_1 i_{L1} - i_{dc1} \tag{7}$$

Looking from the DC-DG2 side, the $v_{dc}$ state is given in Equation (8).

$$C_{d2}\frac{d}{dt}v_{dc} = u_2 i_{L2} - i_{dc2} \tag{8}$$

Looking from the AC side, the $v_{dc}$ state is represented as in Equation (9).

$$C_{dc}\frac{d}{dt}v_{dc} = -u_d i_d - u_q i_q - i_{dc} \tag{9}$$

Choosing $C_{d1}$ and $C_{d2}$ to be equal to $C_{dc}$ and adding Equations (7)–(9) yields:

$$3C_{dc}\frac{d}{dt}v_{dc} = i_{L1}(1 - u_1) + u_2 i_{L2} - u_d i_d - u_q i_q - (i_{dc} + i_{dc1} + i_{dc2}) \tag{10}$$

Equations (2)–(6) and (10) provide the hybrid MG's model, with the states.

$$\mathbf{x} = \begin{bmatrix} i_d & i_q & v_{pv} & i_{L1} & i_{L2} & v_{dc} \end{bmatrix}'$$

The model is nonlinear due to the product of the switching signals and their associated variables. To design the controller, this model should be linearized around an operating point. Here, the operating point is assumed as the voltages and currents' nominal values. Then a perturbation is allowed around such an operating point. For instance, $i_d$ is considered as: $i_d = I_d + \Delta i_d$. $I_d$ is the $i_d$'s value at the operating point (a quiescent value), and $\Delta i_d$ is the perturbation.

*Linearized Model*

Substituting each state by its sum of a quiescent value and a perturbation in the Equations from (2) to (6), and (10), and neglecting the 2nd order terms, we get:

$$
\begin{aligned}
\Delta \dot{i}_d &= \frac{\left(-R_{ac}\Delta i_d + \omega L_{ac}\Delta i_q - \Delta v_d + U_d\Delta v_{dc} + V_{dc}\Delta u_d\right)}{L_{ac}} \\
\Delta \dot{i}_q &= \frac{\left(-\omega L_{ac}\Delta i_d - R_{ac}\Delta i_q - \Delta v_q + U_q\Delta v_{dc} + V_{dc}\Delta u_q\right)}{L_{ac}} \\
\Delta \dot{v}_{pv} &= \frac{\left(-\Delta i_{L1} + \Delta i_{pv}\right)}{C_s} \\
\Delta \dot{i}_{L1} &= \frac{\left(-\Delta v_{dc}(1 - U_1) + v_{dc}\Delta u_1 + \Delta v_{pv}\right)}{L_1} \\
\Delta \dot{i}_{L1} &= \frac{\left(-U_2\Delta v_{dc} - V_{dc}\Delta u_2 + v_{bat}\right)}{L_2} \\
\Delta \dot{v}_{dc} &= \begin{pmatrix} (1 - U_1)\Delta i_{L1} + U_2\Delta i_{L2} - I_{L1}\Delta u_1 + I_{L2}\Delta u_2 - U_d\Delta i_d - I_d\Delta u_d \\ -U_q\Delta i_q - I_q\Delta u_q - \frac{(\Delta i_{dc} + \Delta i_{dc1} + \Delta i_{dc2})}{(3C_{dc})} \end{pmatrix}
\end{aligned}
\tag{11}
$$

where $\Delta \dot{x}$ denotes the first derivative of x, and capital letters designate quiescent values. Thus, in state-space form, the model in Equation (11) is represented as in Equation (12).

$$
\begin{aligned}
\dot{\Delta x} &= A\Delta x + B\Delta u + Dw \\
\Delta y &= C\Delta x \\
\Delta x &= \begin{bmatrix} \Delta i_d & \Delta i_q & \Delta v_{pv} & \Delta i_{L1} & \Delta i_{L2} & \Delta v_{dc} \end{bmatrix}' \\
\Delta u &= \begin{bmatrix} \Delta u_d & \Delta u_q & \Delta u_1 & \Delta u_2 \end{bmatrix}' \\
w &= \begin{bmatrix} \Delta v_d & \Delta v_q & \Delta i_{pv} & \Delta v_{bat} & \Delta i_{dc} & \Delta i_{dc1} & \Delta i_{dc2} \end{bmatrix}' \\
\Delta y &= \begin{bmatrix} \Delta i_d & \Delta i_q & \Delta v_{pv} & \Delta v_{dc} \end{bmatrix}'
\end{aligned}
\tag{12}
$$

where $x \in \mathbb{R}^n$, $u \in \mathbb{R}^p$, $y \in \mathbb{R}^l$, and $w \in \mathbb{R}^m$ are the state, control, output for feedback, and external disturbance vectors, respectively. Also,

$$
A = \begin{bmatrix}
-\frac{R_{ac}}{L_{ac}} & \omega & 0 & 0 & 0 & \frac{U_d}{L_{ac}} \\
-\omega & -\frac{R_{ac}}{L_{ac}} & 0 & 0 & 0 & \frac{U_q}{L_{ac}} \\
0 & 0 & 0 & -\frac{1}{C_s} & 0 & 0 \\
0 & 0 & \frac{1}{L_1} & 0 & 0 & -\frac{(1-U_1)}{L_1} \\
0 & 0 & 0 & 0 & 0 & -\frac{U_2}{L_2} \\
-\frac{U_d}{3\,C_d} & -\frac{U_q}{3\,C_d} & 0 & \frac{1-U_1}{3\,C_d} & \frac{U_2}{3\,C_d} & 0
\end{bmatrix}
$$

$$
B = \begin{bmatrix}
\frac{V_{dc}}{L_{ac}} & 0 & 0 & 0 \\
0 & \frac{V_{dc}}{L_{ac}} & 0 & 0 \\
0 & 0 & 0 & 0 \\
0 & 0 & \frac{V_{dc}}{L_1} & 0 \\
0 & 0 & 0 & -\frac{V_{dc}}{L_2} \\
-\frac{I_d}{3\,C_{dc}} & -\frac{I_q}{3\,C_{dc}} & -\frac{I_{L1}}{3\,C_{dc}} & \frac{I_{L2}}{3\,C_{dc}}
\end{bmatrix}
$$

$$
D = \begin{bmatrix}
-\frac{1}{L_{ac}} & 0 & 0 & 0 & 0 & 0 & 0 \\
0 & -\frac{1}{L_{ac}} & 0 & 0 & 0 & 0 & 0 \\
0 & 0 & \frac{1}{C_s} & 0 & 0 & 0 & 0 \\
0 & 0 & 0 & 0 & 0 & 0 & 0 \\
0 & 0 & 0 & \frac{1}{L_2} & 0 & 0 & 0 \\
0 & 0 & 0 & 0 & -\frac{1}{3C_{dc}} & -\frac{1}{3C_{dc}} & -\frac{1}{3C_{dc}}
\end{bmatrix}
$$

$$
C = \begin{bmatrix}
1 & 0 & 0 & 0 & 0 & 0 \\
0 & 1 & 0 & 0 & 0 & 0 \\
0 & 0 & 1 & 0 & 0 & 0 \\
0 & 0 & 0 & 0 & 0 & 1
\end{bmatrix}
$$

To find the operating point, all the derivatives in equations from Equations (2)–(6) and (10) are set to zero. Then, by substituting all the variables by their values, a function of nine unknowns is formed as in Equation (13). Here is an example to illustrate the steps of obtaining Equation (13). Going back to Equation (2).

$$
L_{ac} \frac{d}{dt} i_d = -R_{ac} i_d + \omega L_{ac} i_q - v_d + u_d v_{dc}
$$

Setting the derivative term to zero gives.

$$
0 = -R_{ac} i_d + \omega L_{ac} i_q - v_d + u_d v_{dc}
$$

Then $i_d \rightarrow I_d$, $i_q \rightarrow I_q$, $v_d \rightarrow V_d$, $u_d \rightarrow U_d$, and $v_{dc} \rightarrow V_{dc}$, yield: $-R_{ac} I_d + \omega L I_q - V_d + U_d V_{dc} = 0$. This is the first row in Equation (13).

$$
f(x) = \begin{bmatrix}
-R_{ac} I_d + \omega L I_q - V_d + U_d V_{dc} \\
-R_{ac} I_q - \omega L I_d - V_d + U_q V_{dc} \\
-I_{L1} + I_{pv} \\
-V_{dc}(1 - U_1) + V_{pv} \\
-U_2 V_{dc} + V_{bat} \\
I_{L1} - U_d I_d - U_q I_q - U_1 I_{L1} + U_2 I_{L2} + I_{dc} + I_{dc1} + I_{dc2}
\end{bmatrix} = 0 \qquad (13)
$$

The knowns are $I_q$, $I_{dc}$, $U_d$, $U_q$, $U_1$, $U_2$, $I_{L1}$, $I_{L2}$, and $V_{dc}$. The parameters in (13) are listed in Tables 1 and 2, with their used values.

**Table 1.** Hybrid MG parameters [19].

| Parameter | Symbol | Value | Unit |
|---|---|---|---|
| AC-side Resistance | $R_{ac}$ | 0.30 | $\Omega$ |
| AC-side Inductance | $L_{ac}$ | 3.0 | mH |
| Input Capacitance of DC-DG1 | $C_s$ | 110.0 | $\mu$F |
| DC-side Capacitance of DC-DG1 | $C_{dc}$ | 60.0 | $\mu$F |
| Inductance of DC-DG1 | $L_1$ | 8.0 | mH |
| Inductance of DC-DG2 | $L_2$ | 8.0 | mH |

**Table 2.** Hybrid MG nominal values [19].

| Variable | Symbol | Value | Unit |
|---|---|---|---|
| Fundamental Frequency | $\omega$ | $100\pi$ | rad/s |
| AC-side Voltage | $V_a$ | 220.0 | V |
| DC voltage of DC-DG1 and DC-DG2 | $V_{pv}, V_{bat}$ | 600.0 | V |
| DC-link Voltage | $V_{dc}$ | 700.0 | V |
| Battery Capacity, DC-DG2 | $C_{bat}$ | 22.0 | Ah |
| AC-side Active Power | $P_{ac}$ | 12.0 | kW |
| AC-side Reactive Power | $Q_{ac}$ | 10.0 | kVAr |
| MPPT PV power | $P_{pv}$ | 4.50 | kW |
| Switching Frequency | $f_{sw}$ | 10.0 | kHz |

In Equation (13), it is noticed that the number of equations is less than the number of unknowns. Thus, the damped least square (Levenberg–Marquardt algorithm, via the function fsolve in Matlab) is used to solve it with an initial guess:

$$x' = [0, 18, 1, 1, 1, 1, 7.5, 7.5, 700].$$

Table 3 shows the solution results for the nine unknowns in (13), noting that the values of $U_1$, $U_2$, $U_d$, $U_q$ are dimensionless.

**Table 3.** Operating point: solution of Equation (13).

| Variable | Value |
|---|---|
| $I_q$ | $-6.7239$ A |
| $I_{dc}$ | 5.3584 A |
| $U_d$ | 0.3311 |
| $U_q$ | 0.0216 |
| $U_1$ | 0.1429 |
| $U_2$ | 0.8571 |
| $I_{L1}$ | 6.0775 A |
| $I_{L2}$ | 19.3244 A |
| $V_{dc}$ | 700.0219 V |

After getting the operation point, these values are substituted in Equation (12) to obtain the linearized hybrid MG model.

The problem can be stated as follows. It is required to design a controller based on the linearized model for the studied hybrid MG. The controller has to be robust against

parameter variations modelled as norm-bounded uncertainty. The proposed controller is to be tested on the original non-linear model of the hybrid MG.

### 3. Controller Design

#### 3.1. The Invariant (or Attractive) Ellipsoid Set

The attractive (or invariant) ellipsoid design has recently been developed in robust control of linear and non-linear systems [22]. Moreover, the ellipsoidal design is successfully applied in many applications, e.g., automatic generation control [23], piezoelectric actuators [24], car active suspension [25], islanded AC microgrids [26,27].

The basic idea of the ellipsoidal design is as follows. For an initial state vector outside the ellipsoid, the state trajectory must be attracted to a small ellipsoidal region, including the origin (attracting ellipsoid). When the trajectory reaches the ellipsoid, it will not leave it (invariant ellipsoid) [28]. The linear time-invariant (LTI) system (12) is rewritten, removing the $\Delta$ for simplicity, as follows:

$$\dot{x} = Ax + Bu + Dw, \quad y = Cx,$$
$$z = C_z x + B_2 u \tag{14}$$

The vector z is the output to be optimized (the effect of disturbance w on z must be minimized). The term $B_2 u$ is added to z to avoid obtaining controller with large gains, challenging to implement in practice. It is assumed that $C_z = C$.

#### 3.2. Design Procedure

There are two control problems: (1) regulator (for constant reference), (2) tracking (for time-varying reference). The output must track the desired time-varying reference with zero steady-state error in the hybrid microgrid control. Therefore, an integrator must be inserted, as will be seen later.

#### 3.3. Regulator Ellipsoidal Design

The LTI system (14) assumes that the pair (A, B) is controllable, and the pair (A, C) is observable under a bounded external disturbance Dw. The disturbance is subject to the following constraint:

$$\|w(t)\| \leq 1, \quad t \geq 0 \tag{15}$$

Note that the constraint in Equation (15) can be realized by scaling the matrix D.

Consider an ellipsoid *E* with origin as the centre given by the following equation:

$$E = x'P^{-1}x \leq 1, \quad P > 0 \tag{16}$$

The ellipsoid also represents a Lyapunov function $V(x) = x'P^{-1}x$, if the initial state vector $x(0)$ lies outside the ellipsoid, V does not increase outside this ellipsoid. In other words, $\dot{V}(x) \leq 0$ for all $x(t)$ subject to $V(x) \geq 1$, i.e., the state trajectory $x(t)$ is attracted to the ellipsoid. When the trajectory of $x(t)$ reaches the ellipsoid, it will not leave it for the future time (time-invariant). The ellipsoid is thus attractive and invariant.

**Remark 2.** *For an initial state vector x(0) starting outside the ellipsoid* $V(x) = x'P^{-1}x$, *the proposed design is based on attracting the state trajectories to the ellipsoid by requiring* $\dot{V}(x) \leq 0$ *for all* $x(t)$ *subject to* $V(x) \geq 1$. *Once the trajectories x(t) arrive at the ellipsoid, it will not leave it for the future time. This constraint and the disturbance norm constraint Equation (15) are cast in one matrix inequality Equation (17) using Schur complements, Fact 2.*

It is shown in [28] that *E* is invariant (and attracting) for the system (14) if and only if there exists a scalar $\alpha > 0$, and a symmetric matrix $P > 0$ solutions to the following non-linear matrix equation:

$$\begin{bmatrix} (AP + BY + *) + \alpha P & D \\ * & -\alpha I \end{bmatrix} \leq 0 \tag{17}$$

where $Y = KP$. The source of non-linearity is the product term $\alpha P$. The disturbance effect on the trajectory x can be rejected by minimizing the ellipsoid volume as follows:

$$minimize\ tr(P) \tag{18}$$

where $tr(P)$ is the trace function (the sum of the diagonal elements of P), selected due to its linearity, which provides a semidefinite program (SDP) that can be easily solved by Matlab convex optimization, as will be seen later.

To find the effect of the disturbance on the output to optimize $z = (C + B_2K)x$, the ellipsoid in Equation (16) is replaced by:

$$E_z = z'((C + B_2K)P(C + B_2K)')^{-1}z \leq 1 \tag{19}$$

The effect of the disturbance on z is minimized by:

$$minimize\ tr\left[(C + B_2K)P(C + B_2K)'\right] \tag{20}$$

The proposed controller, which reduces (or rejects) the disturbance effect on the output **z**, is obtained by the following theorem.

**Theorem 1** [28]**.** *Consider the LTI system (14), subject to the disturbance described in (15). Then, a state feedback regulator can stabilize the system:*

$$u = Kx \tag{21}$$

*is equivalent to:*

$$minimize\ tr\left[CPC' + (CY'B_2' + *) + B_2ZB_2'\right] \tag{22}$$

*subject to the constraints:*

$$\begin{bmatrix} (AP + BY + *) + \alpha P & D \\ * & -\alpha I \end{bmatrix} \leq 0, \alpha > 0$$
$$\begin{bmatrix} Z & Y \\ * & P \end{bmatrix} \geq 0,\ P > 0 \tag{23}$$

*where $Y = KP$ and the minimization is carried out to the variables $\alpha$, $P = P' \in \mathbb{R}^{n \times n}, Y \in \mathbb{R}^{p \times n}$ and $Z = Z' \in \mathbb{R}^{p \times p}$. Let $Y = KP$. Then the regulator is given by the Equation (24).*

$$K = YP^{-1} \tag{24}$$

*Fixing $\alpha$ in the nonlinear matrix Equation (23) results in an LMI. The objective function is iteratively minimized by fixing $\alpha$ for every iteration. Therefore SDP in one-dimensional convex optimization is obtained and easy to solve by the LMI toolbox.*

*3.4. Tracker Ellipsoidal Design*

The above theorem can be generalized to the tracking problem by inserting an integral part between the plant's error signal and the plant, as shown in Figure 5.

State feedback plus integral control is proposed in the form:

$$u = K x + K_i \, \xi = \begin{bmatrix} K & K_i \end{bmatrix} \begin{bmatrix} x \\ \xi \end{bmatrix}, \; \dot{\xi} = r - C x \tag{25}$$

Therefore, the augmented system is:

$$\begin{bmatrix} \dot{x} \\ \dot{\xi} \end{bmatrix} = \begin{bmatrix} A & 0 \\ -C & 0 \end{bmatrix} \begin{bmatrix} x \\ \xi \end{bmatrix} + \begin{bmatrix} B \\ 0 \end{bmatrix} u + \begin{bmatrix} D \\ 0 \end{bmatrix} w + \begin{bmatrix} 0 \\ I \end{bmatrix} r, \; z = \begin{bmatrix} C & 0 \end{bmatrix} \begin{bmatrix} x \\ \xi \end{bmatrix} + B_2 u \tag{26}$$

or

$$\begin{bmatrix} \dot{x} \\ \dot{\xi} \end{bmatrix} = \hat{A} \begin{bmatrix} x \\ \xi \end{bmatrix} + \hat{B} \, u + \hat{D} \, w + \begin{bmatrix} 0 \\ I \end{bmatrix} r, z = \hat{C} \begin{bmatrix} x \\ \xi \end{bmatrix} + B_2 u \tag{27}$$

where the augmented matrices are:

$$\hat{A} = \begin{bmatrix} A & 0 \\ -C & 0 \end{bmatrix}, \hat{B} = \begin{bmatrix} B \\ 0 \end{bmatrix}, \hat{D} = \begin{bmatrix} D \\ 0 \end{bmatrix} \quad \hat{C} = \begin{bmatrix} C \\ 0 \end{bmatrix} \tag{28}$$

The design of regulators via the ellipsoidal sets, Theorem 1, can be generalized to the problem of the tracker's design. The suggested controller must be robust against the parameters uncertainties and load variations of the hybrid MG. Thus, Equation (28) is adapted to:

$$\begin{bmatrix} \dot{x} \\ \dot{\xi} \end{bmatrix} = (\hat{A} + \Delta\hat{A}) \begin{bmatrix} x \\ \xi \end{bmatrix} + (\hat{B} + \Delta\hat{B})u + (\hat{D} + \Delta\hat{D})w + \begin{bmatrix} 0 \\ I \end{bmatrix} r, \; z = \hat{C} \begin{bmatrix} x \\ \xi \end{bmatrix} + B_2 u \tag{29}$$

where the following norm-bounded model represents the uncertainties:

$$\begin{aligned} \Delta\hat{A} &\leq M\Delta(t)N, \; \|\Delta(t)\| \leq 1; \; \Delta\hat{B} \leq M_B\Delta_B(t)N_B, \; \|\Delta_B(t)\| \leq 1 \\ \Delta\hat{D} &\leq M_D\Delta_D(t)N_D, \; \|\Delta_D(t)\| \leq 1 \end{aligned} \tag{30}$$

The norm-bounded form can be obtained using the singular value decomposition (decomposing a matrix into the multiplication of three matrices). Finally, Theorem 2 is developed to determine the controller without any trial and error.

**Theorem 2.** *Assume that the system (29) is controllable* $(\hat{A}, \hat{B})$, *and observable* $(\hat{A}, \hat{C})$, *under $L_\infty$-bounded exogenous disturbances. Then, the robust disturbance-rejection state feedback plus integral controller* $u = \begin{bmatrix} K & K_i \end{bmatrix} \begin{bmatrix} x \\ \xi \end{bmatrix}$ *is obtained by solving: minimize* $tr\left[\hat{C}\hat{P}\hat{C}' + \left(\hat{C}\hat{Y}'B_2' + *\right) + B_2 Z B_2'\right]$, *subject to the following constraints:*

$$\begin{bmatrix} \varphi & \hat{D} & \hat{P}N' & 0 & \hat{Y}'N_B' \\ * & -\alpha I & 0 & N_D' & 0 \\ * & * & -\epsilon I & 0 & 0 \\ * & * & * & -\rho I & 0 \\ * & * & * & * & -\pi I \end{bmatrix} \leq 0, \quad \alpha, \epsilon, \rho, \pi > 0 \tag{31}$$

Where $\varphi = (\hat{A}\hat{P} + \hat{B}\hat{Y} + *) + \alpha\hat{P} + \epsilon MM' + \rho M_D M_D' + \mu M_B M_B'$

$$\begin{bmatrix} Z & \hat{Y} \\ * & \hat{P} \end{bmatrix} \geq 0, \; \hat{P} > 0$$

*where* $\hat{Y} = \hat{K}\hat{P}, \hat{K} = \begin{bmatrix} K & K_i \end{bmatrix}$. *The minimization is carried out to the variables* $\dots \varepsilon, \rho, \mu$, $P = P' \in \mathbb{R}^{(n+1)\cdot(n+1)}, \hat{Y} \in \mathbb{R}^{p\cdot(n+1)}$. *Note that this theorem does not impose the matching condition* $\begin{bmatrix} \Delta\hat{A} & \Delta\hat{B} & \Delta\hat{D} \end{bmatrix} = M\Delta(t)\begin{bmatrix} N & N_B & N_D \end{bmatrix}$.

**Proof.** For the tracking problem, the matrices in (14) are replaced by the augmented system matrices. Thus, Equation (17) becomes:

$$\begin{bmatrix} (\hat{A}\hat{P} + \hat{B}\hat{Y} + *) + \alpha\hat{P} & \hat{D} \\ * & -\alpha I \end{bmatrix} \le 0, \alpha > 0 \tag{32}$$

To tackle the load variations in the hybrid MG, $\hat{A}$ is substituted by $(\hat{A} + \Delta\hat{A})$, $\hat{B}$ by $(\hat{B} + \Delta\hat{B})$, and $\hat{D}$ by $(\hat{D} + \Delta\hat{D})$ in Equation (32), resulting in:

$$\begin{bmatrix} ((\hat{A} + \Delta\hat{A})\hat{P} + (\hat{B} + \Delta\hat{B})\hat{Y} + *) + \alpha\hat{P} & \hat{D} + \Delta\hat{D} \\ * & -\alpha I \end{bmatrix} \le 0, \ \alpha > 0$$

Substituting for the norm-bounded uncertainty Equation (30), one obtains:

$$\begin{bmatrix} ((\hat{A} + M\Delta N)\hat{P} + (\hat{B} + M_B\Delta_B N_B)\hat{Y} + *) + \alpha\hat{P} & \hat{D} + M_D\Delta_D N_D \\ * & -\alpha I \end{bmatrix} \le 0, \ \alpha > 0$$

Separating the uncertainty terms, we obtain:

$$\begin{bmatrix} (\hat{A}\hat{P} + \hat{B}\hat{Y} + *) + \alpha\hat{P} & \hat{D} \\ * & -\alpha I \end{bmatrix} + \left( \begin{bmatrix} M \\ 0 \end{bmatrix} \Delta \begin{bmatrix} N\hat{P} & 0 \end{bmatrix} + * \right) \\ + \left( \begin{bmatrix} M_B \\ 0 \end{bmatrix} \Delta_B \begin{bmatrix} N_B\hat{Y} & 0 \end{bmatrix} + * \right) + \left( \begin{bmatrix} M_D \\ 0 \end{bmatrix} \Delta_D \begin{bmatrix} 0 & N_D\hat{Y} \end{bmatrix} + * \right) \le 0 \tag{33}$$

Using Fact 1 to eliminate $\Delta_A(t)$, $\Delta_B(t)$, and $\Delta_D(t)$, (33) is fulfilled if the following matrix equation is satisfied:

$$\begin{bmatrix} (\hat{A}\hat{P} + \hat{B}\hat{Y} + *) + \alpha\hat{P} & \hat{D} \\ * & -\alpha I \end{bmatrix} + \epsilon \begin{bmatrix} M \\ 0 \end{bmatrix} \begin{bmatrix} M \\ 0 \end{bmatrix}' + \epsilon^{-1} \begin{bmatrix} \hat{P}N' \\ 0 \end{bmatrix} \begin{bmatrix} \hat{P}N' \\ 0 \end{bmatrix}'$$

$$+ \mu \begin{bmatrix} M_B \\ 0 \end{bmatrix} \begin{bmatrix} M_B \\ 0 \end{bmatrix}' + \mu^{-1} \begin{bmatrix} \hat{Y}'N'_B \\ 0 \end{bmatrix} \begin{bmatrix} \hat{Y}'N'_B \\ 0 \end{bmatrix}' + \rho \begin{bmatrix} M_D \\ 0 \end{bmatrix} \begin{bmatrix} M_D \\ 0 \end{bmatrix}' + \rho^{-1} \begin{bmatrix} 0 \\ N'_D \end{bmatrix} \begin{bmatrix} 0 \\ N'_D \end{bmatrix}' \le 0, \quad \alpha, \epsilon, \rho, \mu > 0$$

Using Fact 2, Theorem 2 is proved. □

### 3.5. Gain Matrices

Solving the above LMI (31) yields the following gain matrices of the proposed controller.

$$\mathbf{K} = \begin{bmatrix} -905.75, & -0.0031213 & 0.45549 & 0.74613 & -0.35706 & 1623.9 \\ -0.22601 & -906.7 & -0.21299 & -0.22843 & 0.1109 & -504.42 \\ 116.63 & 1.3025 & 2819.2 & 97.073 & -59.051 & 2.6869 \times 10^5 \\ 1096.1 & 2.2905 & 879.07 & 1178.7 & -559.74 & 2.5753 \times 10^6 \end{bmatrix}$$

$$\mathbf{K_i} = \begin{bmatrix} -116.8 & 127.98 & 121.12 & 37.791 \\ 67.158 & -306.96 & -277.58 & -188.36 \\ -242.35 & 985.24 & 341.2 & -341.89 \\ 35{,}952 & 15{,}082 & -7324.5 & 15{,}313 \end{bmatrix}$$

## 4. Simulation Results

The hybrid MG system shown in Figures 3 and 4 was studied, tested, and implemented using Matlab/Simulink Power Library. The ellipsoid tracking control algorithm was assessed with two different sets of the MG parameters provided in Table 4. Table 2 provides further specifications of the proposed MG.

**Table 4.** Two different sets of MG parameters.

| Parameter | First Set | Second Set |
|---|---|---|
| AC-side Resistance ($R_{ac}$) | 100% | 115% |
| AC-side Inductance ($L_{ac}$) | 100% | 90% |
| Input Capacitance of DC-DG1 ($C_s$) | 100% | 85% |
| DC-side Equivalent Capacitance of DC-DG1 ($C_{dc}$) | 100% | 110% |
| Inductance of DC-DG1 ($L_1$) | 100% | 85% |
| Inductance of DC-DG2 ($L_2$) | 100% | 85% |

The DGs on the AC MG provide their local load demands. When the CB1 is closed, the load connected to the AC-DG1 (wind turbine system) is changed at $t$ = 0.8 s, increased by 1.25 kW and 1.0 kVAR. Moreover, when the CB2 is closed, the load connected to AC-DG2 (PV inverter 2) is changed at $t$ = 1.2 s, increased by 2.2 kW and 1.5 kVAR. Both cases will happen under two different sets of the MG parameters, as illustrated in Table 4. The proposed control technique was compared with the sliding mode controller used in [19]. The comparison between both techniques is shown in all figures from Figures 6–10 and Figures 12–17.

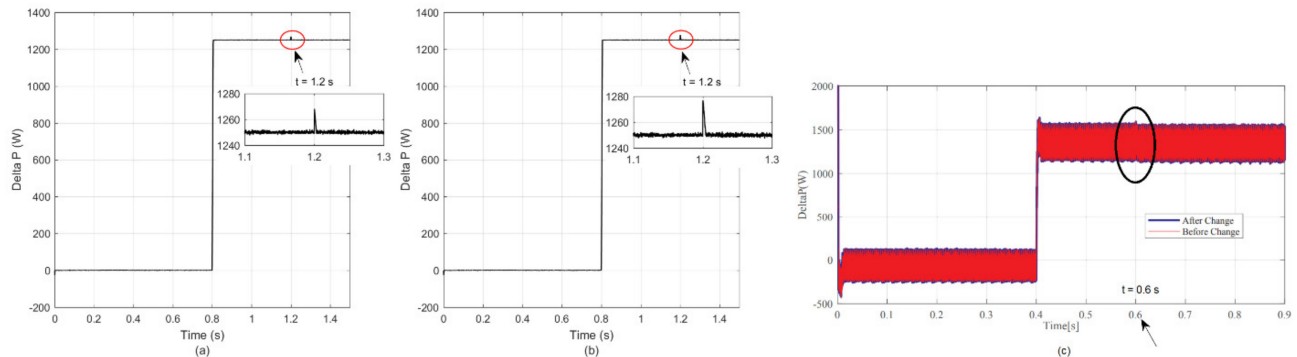

**Figure 6.** Variation of the active power of AC-MG compensated by inverter 2, (**a**) with the first set of MG parameters, (**b**) with the second set of MG parameters, and (**c**) with two various sets of MG parameters in reference [19].

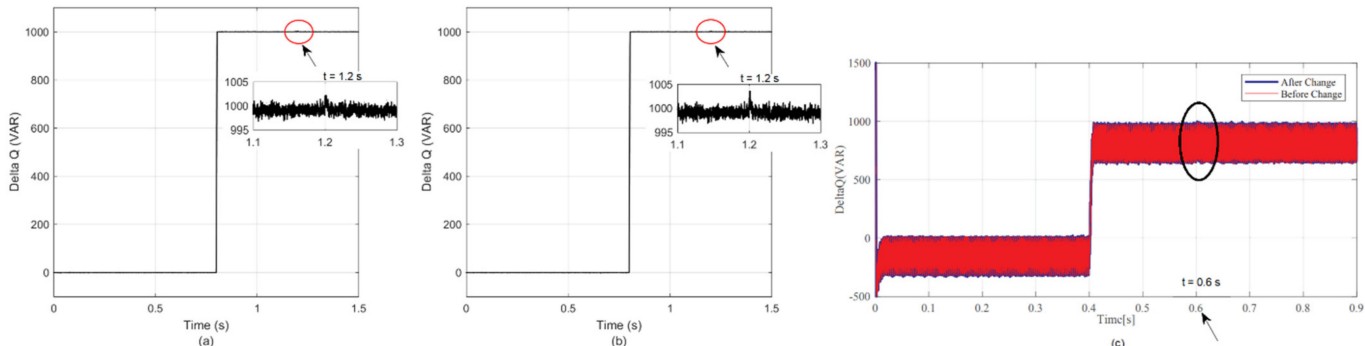

**Figure 7.** Variation of reactive power of AC-MG compensated by inverter 2, (**a**) with the first set of MG parameters, (**b**) with the second set of MG parameters, and (**c**) with two different sets of hybrid MG parameters in reference [19].

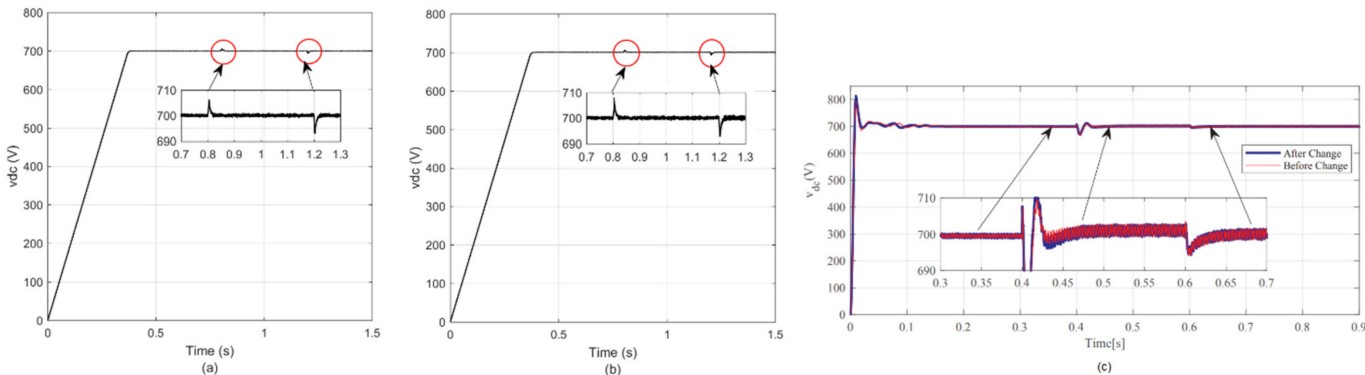

**Figure 8.** The DC-link voltage under two different sets of hybrid MG parameters, (**a**) under the first set, (**b**) under the second set, (**c**) under two different sets of MG parameters in ref [19].

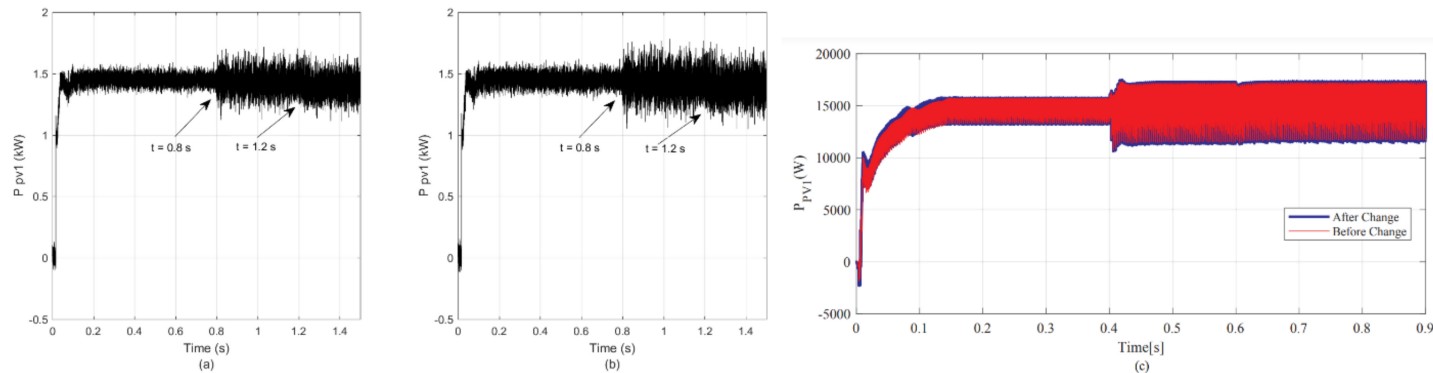

**Figure 9.** Injected power by DG-DC1 (PV1) with two different sets of MG parameters, (**a**) with the first set, (**b**) under the second set, (**c**) with two different sets of MG parameters in ref [19].

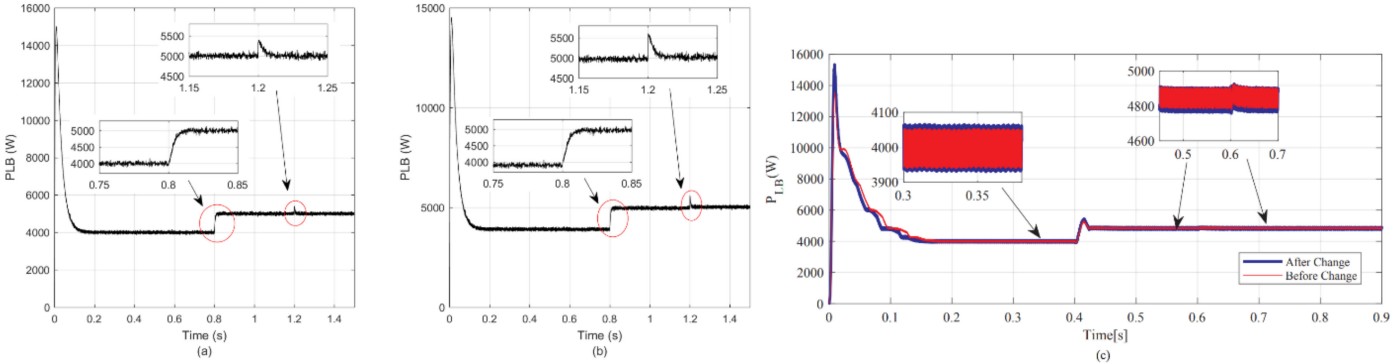

**Figure 10.** Injected power by DGDC2 (lithium battery) with two different parameter sets of MG, (**a**) with the first set, (**b**) with the second set, (**c**) under two different sets of MG parameters in ref [19].

The changes of power (active and reactive) in the AC MG are given in Figures 6 and 7. At $t = 0.8$ s, changes in active power (Delta P) are shown in Figures 6a and 7a for the first and second parameters sets, respectively. The changes in reactive power (Delta Q) are displayed in Figures 6b and 7b for the first and second parameter sets, respectively. Furthermore, Figures 6c and 7c show the sliding mode technique [19] results for this case. The proposed technique is fast, with low spikes, and has no chattering compared to the sliding mode technique used in [19].

The transient and steady-state responses of the DC, AC-MG state variables, and power-sharing of the hybrid MG using the proposed technique is tested, verified, and compared to the sliding mode technique used in [19] in the following subsections.

### 4.1. Direct Current (DC) Side State Variables

The DC-link voltage of the inter-linking converter is illustrated in Figure 8. In Figure 8a,b, the DC-link as a dc state variable is assessed with the two different sets of the MG parameters (Table 4). The DC-link voltage responses with these two sets of parameters and added loads in the AC side during $t = 0.8$ s and $t = 1.2$ s respond perfectly with a difference between the spikes rates that is too small. Therefore, $v_{dc}$ with a stable steady-state and dynamic response is accomplished. Figure 8c shows the DC-link voltage response for the sliding mode control technique [19]. The comparison between both techniques shows that the DC-link voltage response's proposed technique is a bit slower. Nonetheless, it rejects the load disturbance efficiently without the high chattering in the sliding-mode control technique [19].

The output power of the DC/DC converter of the DC-DG2 and the bidirectional buck-boost DC/DC converter of the DC-DG2 (LB) are shown in Figures 9 and 10, respectively. Both figures prove that the proposed technique effectively forces both converters to partially generate the power demanded during AC-side load changes.

In Figure 9a,b, the performance of the proposed controller with the two sets of the hybrid MG is depicted, while, in Figure 9c, the performance of the sliding-mode technique used in [19] is illustrated with small fluctuations detected in the output power of DCDG1 when the second set of parameters is applied. In Figure 10a,b, the proposed controller in the DCDG1 (LB) within the two sets of the hybrid MG is operated successfully with a dead-beat response. In Figure 10c, the sliding-mode technique used in [19] run with more significant overshoot and sluggish transient response with the second parameter's set.

**Remark 3.** *The significant reduction of chattering (high-frequency signal resulting from the inverters switching) using the proposed design compared with [19] is due to utilizing the integral control in the outer loop. The swiftness and external disturbance attenuation of the proposed control is due to minimizing the ellipsoid's volume.*

### 4.2. Utility Grid State Variables

To test the proposed controller's robustness, we have to evaluate all utility grid state variables' responses. It is recommended to attain a unity-power factor in the utility grid. Figure 11 shows that the a-phase voltage and current are in phase. Both waveforms are sinusoidal, specifying that the power factor between the voltage and current in phase a is approximately unity. In Figures 12 and 13, the utility grid frequency and magnitude of a-phase voltage under two sets of the hybrid MG system are obtained. The proposed control technique is shown in Figures 12a,b and 13a,b can accurately enforce the inter-linking DC/AC power converter to preserve both grid voltage magnitude and frequency stability with an acceptable transient response. In Figures 13c and 14c, the sliding-mode technique is used in [19]. The grid voltages in Figure 13c approach their desired values with more steady and dynamic errors, but they are within an acceptable range for the second set of parameters.

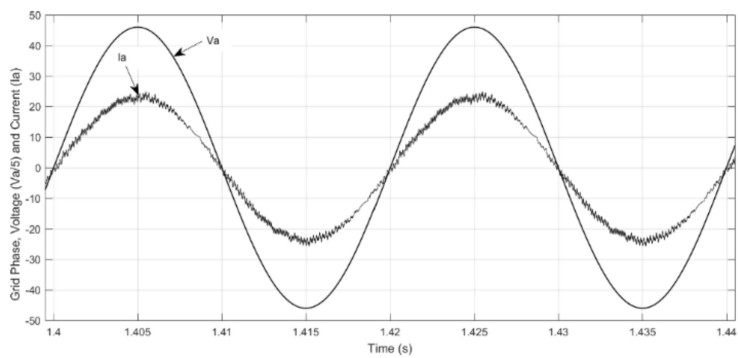

**Figure 11.** The utility grid voltage and current in phase a.

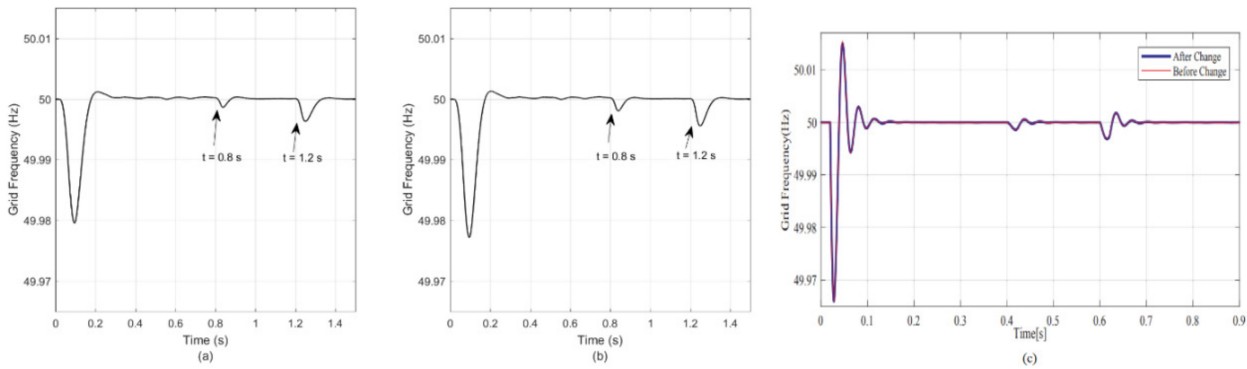

**Figure 12.** Frequency of grid with two different sets of MG parameters, (**a**) under the first set, (**b**) under the second set, (**c**) under two different sets of MG parameters in reference [19].

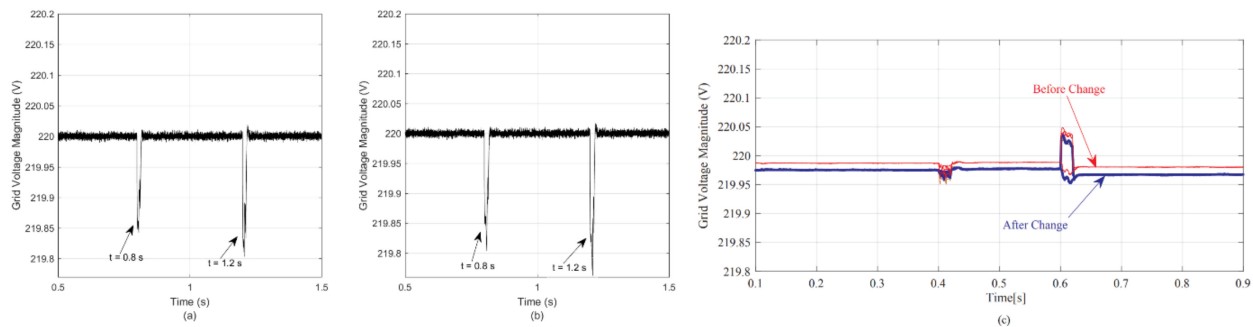

**Figure 13.** The grid voltage magnitudes under two different sets of MG parameters, (**a**) phase a voltage magnitude under the first set, (**b**) phase a voltage magnitude under the second set, (**c**) three-phase voltage magnitudes under two different sets of MG parameters in ref [19].

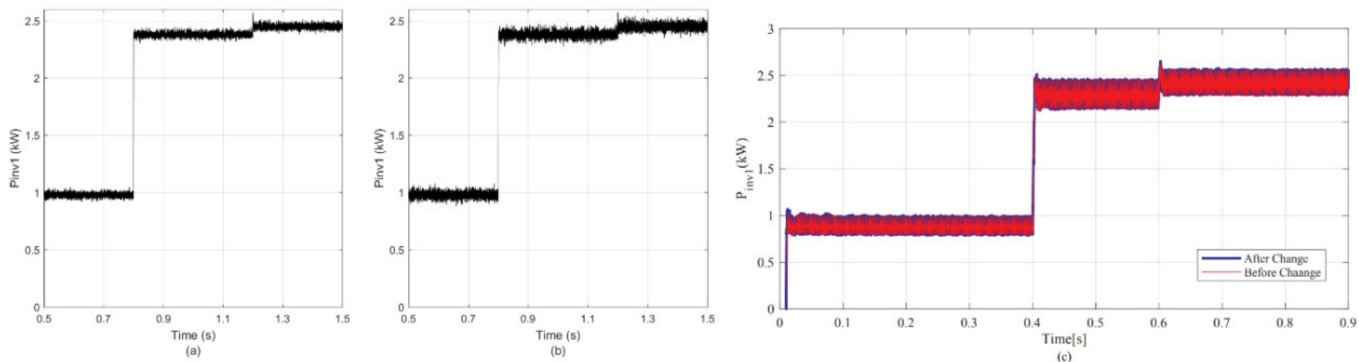

**Figure 14.** Active power of inter-linking inverter (inv1) under two different sets of MG parameters, (**a**) under first set, (**b**) under the second set, (**c**) under two different sets of MG parameters in reference [19].

### 4.3. Power-Sharing between Converters

The inter-linking DC/AC converter must deliver a share of load power and the AC side's uncompensated power, as demonstrated in Figures 14 and 15. Based on these figures, the inter-linking DC/AC converter's power can successfully track its desired value with less steady-state error and a swift dynamic for both sets of hybrid MGs parameters given in Table 4. Figures 14c and 15c show the inter-linking DC/AC converter power-sharing of grid-connected load power for the sliding-mode technique used in [19]. The proposed technique is faster, with a dead-beat response and less chattering.

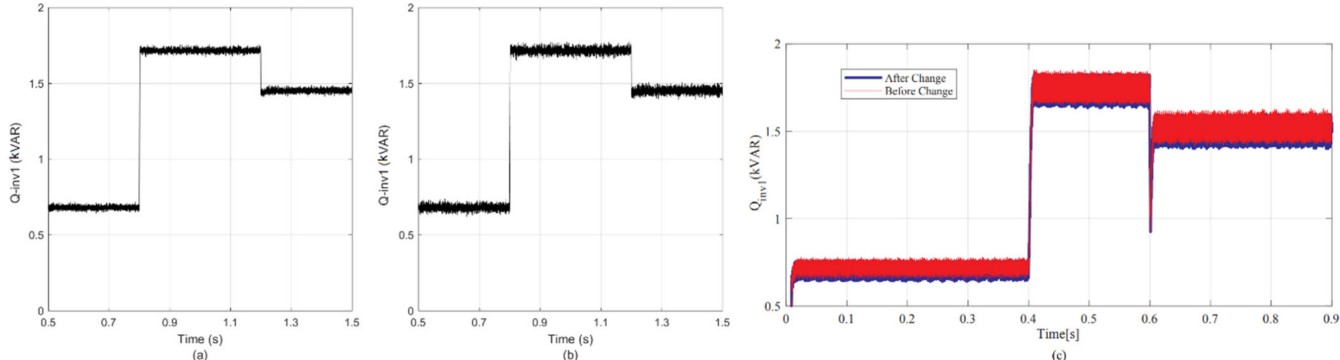

**Figure 15.** Reactive power of inter-linking inverter (inv1) under two different sets of MG parameters, (**a**) under first set, (**b**) under the second set, (**c**) under two different sets of MG parameters in reference [19].

In the AC MG, the PV-inverter 1 is endorsed to provide the active and reactive power necessary for its associated local loads increased at $t$ = 1.2 s. In Figure 16a,b and Figure 17a,b, the proposed control technique can afford stable responses for both the active and reactive power of PV-inverter 2 under both load and parameter changes. While in Figures 16c and 17c, the active and reactive power of PV-inverter 2 with the sliding-mode control used in [19] are shown. The proposed technique is faster, with a dead-beat response and less chattering.

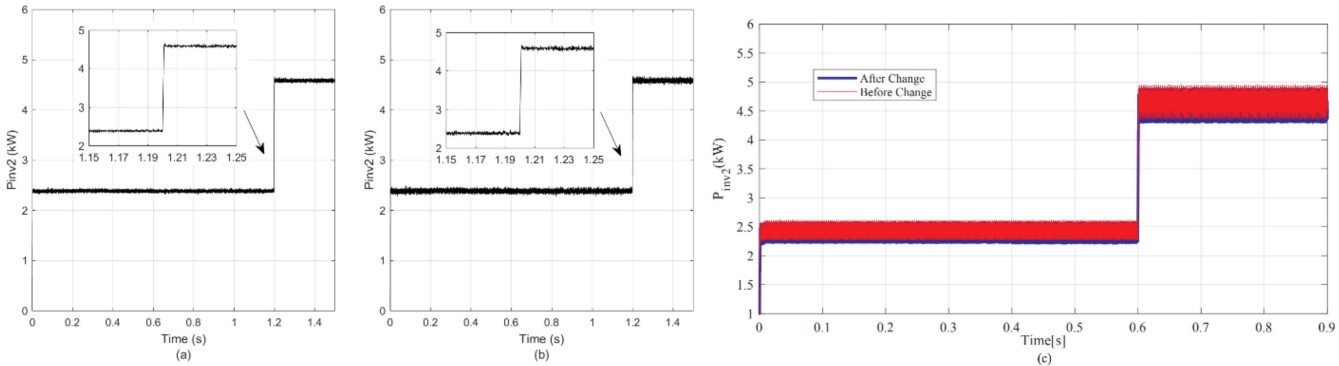

**Figure 16.** Active power of ACDG1 inverter (inv2) under two different sets of MG parameters, (**a**) under first set, (**b**) under the second set, (**c**) under two different sets of MG parameters in reference [19].

Note that the proposed controller based on the linearized model is tested on the original non-linear system in the above simulation. The proposed design is based on satisfying a set of LMIs. It guarantees the stability and disturbance rejection of the linearized model. The stability of the original non-linear system is also guaranteed according to the Lyapunov theorem. This states that the performance of both the linearized and the original non-linear system is the same (both stable or unstable) provided that the linearized model has no eigenvalues on the imaginary axis.

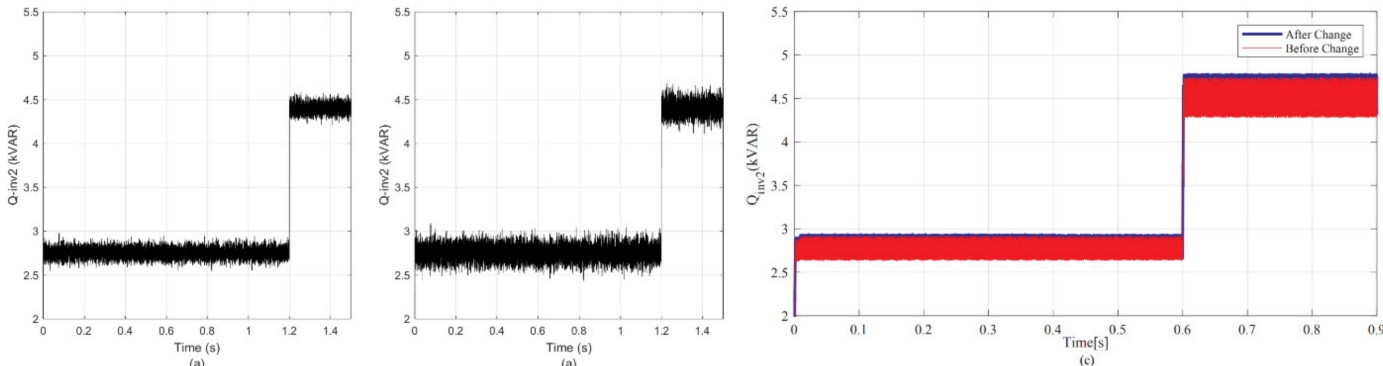

**Figure 17.** Reactive power of ACDG1 inverter (inv2) under two different sets of MG parameters, (**a**) under first set, (**b**) under the second set, (**c**) under two different sets of MG parameters in reference [19].

## 5. Conclusions

The paper has introduced a robust and disturbance-rejection control for hybrid MGs. The hybrid MG model is obtained and linearized. The proposed state feedback plus integral control is based on a novel sufficient condition in terms of LMIs. The design exploited the invariant ellipsoidal set. The MG model is simulated using Matlab/Simulink for different operational scenarios to verify the proposed controller's validity. All the results are compared with a recent controller for the same model in the literature. The results confirm that the proposed algorithm, although it has a marginally slower response to the DC-link voltage, is superior in disturbance rejection, power-sharing, and power-factor correction.

**Author Contributions:** Investigation, Validation, and original draft, H.A.; Methodology, Software, Visualization, Writing, review and editing E.H.E.B.; Methodology, Supervision, Validation, Writing review and editing, H.M.S.; Reviewing and editing, M.D.S. All authors have read and agreed to the published version of the manuscript.

**Funding:** This research received no external funding.

**Conflicts of Interest:** The authors declare no conflict of interest.

## Nomenclature

| Symbol(s) | Interpretation |
| --- | --- |
| $(.)'$ | transpose of a matrix or a vector |
| *, in matrix | symmetric part of the matrix. For instance, the block M + N + * represents the block M + N + M' + N'. |
| 0 | zero matrix |
| $C_{d1}$ | output capacitance of DC-DG1 |
| $C_{d2}$ | output capacitance of DC-DG2 |
| $C_{dc2}$ | dc-link capacitance of DC-DG2 |
| $C_s$ | source capacitance of DC-DG1 |
| DC-DG1 and DC-DG2 | dc-side distributed generators |
| E | n-dimensional ellipsoid |
| I | identity matrix |
| $i_a$, $i_b$, and $i_c$ | instantaneous values of the utility grid current |
| $i_d$ and $i_q$ | d-and the q-components of the grid current |
| $i_{L1}$ and $i_{L2}$ | inductor current of DC-DG1 and DC-DG2, respectively |
| K | state-feedback gain matrix |
| Ki | integral gain matrix |
| $L_1$ and $L_2$ | inductances of DC-DG1 and DC-DG2, respectively |
| P > 0 or P < 0 | symmetric positive definite or negative definite matrix |
| $R_{ac}$ and $L_{ac}$ | resistance and inductance of the ac-side linking filters |

| | |
|---|---|
| $\mathbb{R}^m$ | set of vectors of dimensions m × 1 |
| $\mathbb{R}^{rxq}$ | set of real matrices of dimensions r × q |
| $u_1$ and $u_2$ | switching functions of DC-DG1 and DC-DG2 |
| $u_a$, $u_b$, and $u_c$ | modulation indices for three-phase a, b, and c, respectively |
| $u_d$ and $u_q$ | d-and the q-components of the modulation index |
| V(x) | Lyapunov function |
| $v_a$, $v_b$, and $v_c$ | instantaneous values of the utility grid voltages |
| $v_{dc}$ | dc-side voltage |
| $v_a$, $v_b$, and $v_c$ | instantaneous values of the utility grid voltage |
| $v_d$ and $v_q$ | d-and the q-components of the grid voltage |
| $v_{pv}$ | output voltage of PV plant |
| $v_{pvbat}$ | output voltage of PV plant lithium-ion battery |
| $\omega$ | frequency of the utility grid in rad/s |
| w | disturbance vector |
| x | state vector |
| y | output vector |
| z | accessed/measured states |
| $\alpha$, $\epsilon$, and $\rho$ | scalars |

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
