# Peer review of "Robust Tracker of Hybrid Microgrids by the Invariant-Ellipsoid Set"

_electronics, doi:10.3390/electronics10151794_

Round 1
Reviewer 1 Report
This paper presents a hybrid microgrid control algorithm based on invariant ellipsoids. Simulation results based on MATLAB/Simulink show that the proposed technique is fast with a very low overshoot and features no chattering, unlike an existing sliding mode approach. Despite the clear organization of this work, the following issues need to be further clarified.
1. While the authors have mentioned several existing control algorithms (e.g., LQR [9] and sliding mode [10]) to realize decentralized and centralized MG controllers, the corresponding description is insufficient. I recommend the authors to use a separated section to demonstrate these existing algorithms and explain their differences from the proposed one in this work.
2. While the authors explain each variable denotation used in this work in Table 1, 2, and 3, I recommend the authors to provide a summary of the meanings of all the denotations used in this work before Section 2 so that readers can understand each equation as they read this paper.
3. In Line 280 of Page 13, the authors claim “the proposed technique is fast, with low spikes, has no chattering compared to the sliding 280 mode technique used in [15]”. Then, what are the main reasons that make the technique in [15] show different performances in comparison with the proposed one in this work?
4. There are many writing flaws in this manuscript. The following is an example.
- Line 252 of Page 12: “Solving the above LMI (31) yield ~” should be “Solving the above LMI (31) yields ~”.
Author Response
- While the authors have mentioned several existing control algorithms (e.g., LQR [9] and sliding mode [10]) to realize decentralized and centralized MG controllers, the corresponding description is insufficient. I recommend the authors to use a separated section to demonstrate these existing algorithms and explain their differences from the proposed one in this work.
A new paragraph is added after [18].
Remark 1: It should be emphasized that [14] solves the problem of severe transients that occur during the transition between the connected and islanded modes of microgrids (which affect the voltage and frequency responses). The seamless transition is achieved by the linear quadratic regulators LQR with a simple pole placement method to select the weighting matrices Q and R. The bumpless transition is not considered in the present manuscript. The adaptive sliding control given in [15] achieves fast bus voltage tracking without additional sensors in addition to system scalability and main-taining the plug-and-play operation of the DGs. By using the proposed method [15] to estimate the uncertainty and ex-ternal disturbance, the system parameters and switching function can be adjusted in time to weaken the chattering. However, the control of [15] is nonlinear, has some chattering, and applied to only DC microgrids. The model-predictive control [16] is a time-varying tracker and requires large computational time. The methods [5,6,18] also require extensive training and computational burdens.
- While the authors explain each variable denotation used in this work in Table 1, 2, and 3, I recommend the authors to provide a summary of the meanings of all the denotations used in this work before Section 2 so that readers can understand each equation as they read this paper.
Done
- In Line 280 of Page 13, the authors claim, “the proposed technique is fast, with low spikes, has no chattering compared to the sliding 280 mode technique used in [15]”. Then, what are the main reasons that make the technique in [15] show different performances in comparison with the proposed one in this work?
Remark 2: The significant reduction of chattering (high frequency signal resulting from the inverters switching) using the proposed design as compared with [19] is due to utilizing the integral control in the outer loop. The swift-ness and external-disturbance attenuation of the proposed control is due to minimizing the ellipsoid volume.
- There are many writing flaws in this manuscript. The following is an example. - Line 252 of Page 12: “Solving the above LMI (31) yield ~” should be “Solving the above LMI (31) yields ~”.
Done
Reviewer 2 Report
- More references from mdpi (e.g. Energies) journal or other high quality journals and/or other conferences on hybrid AC and/or DC microgrids/distribution systems.
- Please clarify your innovation/new contribution of this paper, regarding your self-references (recorded in References section).
- The authors are suggested to include a stability analysis, even if on the linearized model of the hybrid system.
- Please use larger font in figures of Simulation section in order the results to be more clear.
Author Response
- More references from mdpi (e.g. Energies) journal or other high quality journals and/or other conferences on hybrid AC and/or DC microgrids/distribution systems.
Three more reference from MDPI Journals were added.
2. Please clarify your innovation/new contribution of this paper, regarding your self-references (recorded in References section).
The new contribution of this paper compared to what we have did before in the recorded references are:
- In our previous work [26. 27], the control idea is applied to only one type of microgrids, the AC microgrid. In this work it is applied to hybrid microgrids in which both types; AC, and DC microgrids are included.
- In our previous work [26, 27] most of Distributed generated (DG) units are PV units. In this work we have many types of DG units as: Wind, batteries and PVs.
- Previously [27], in AC microgrids the DGs are directly connected to each other and to the main grid either. Here, the power flow from DC to AC and vice versa is achieved by through an interlink converter. This adds more difficulty to the problem.
- The authors are suggested to include a stability analysis, even if on the linearized model of the hybrid system.
Note that: In the above simulation the obtained controller is tested in the original nonlinear system. The proposed design is based on satisfying a set of LMIs. It guarantees stability and disturbance rejection of the linearized model. Stability of the original nonlinear system is also guaranteed according to Lyapunov theorem. It states that the performance of both the linearized and the original nonlinear system is the same (both stable or unstable) provided that the linearized model has no eigenvalues on the imaginary axis.
4. Please use larger font in figures of Simulation section in order the results to be more clear.
All Figures were enlarged and they now seem reasonably clear.
Reviewer 3 Report
I have some concerns in this paper:
- The equations and the model in Section 2 are completely described in reference [15]. Why do the authors repeat them here?
- How did you obtain the equation (13) in Section 3?
- Please state that where is your innovation in this paper. Most of the equations and models are used in some other papers. Please clarify.
- Section B.2 (Tracker Ellipsoidal Design) is not clear for me. Please describe more and clear this section.
Author Response
I have some concerns in this paper:
- The equations and the model in Section 2 are completely described in reference [15] [19]. Why do the authors repeat them here?
The same model was chosen for a fair comparison with the sliding mode algorithm. The model was described and rewritten in our paper for the sake of completeness and the reader’s convenience.
2. How did you obtain the equation (13) in Section 3?
Explained in the paper.
3. Please state that where is your innovation in this paper. Most of the equations and models are used in some other papers. Please clarify.
The innovation in this paper regarding paper equations are:
- The nonlinear model in [19] gives partial representation. In the present work the model gives complete representation and linearized.
- The proposed tracker for the hybrid microgrid is totally new.
- The Ellipsoid-based tracker [27] is applied to AC microgrids. In the present work, the approach is extended to the hybrid microgrids. In which the controller elements are changed due to the existence of the interlink converter.
4. Section B.2 (Tracker Ellipsoidal Design) is not clear for me. Please describe more and clear this section.
Remark 3: For an initial state vector x(0) starting outside the ellipsoid , the proposed design is based on attracting the state trajectories to the ellipsoid by requiring Once the trajectories x(t) arrive the ellipsoid, it will not leave it for the future time. This constraint and the disturbance norm constraint (15) are cast in one matrix inequality (17) using Schur complements, Fact 2.